# Solubility and Crystallization Studies of Picolinic Acid

Diogo S. Baptista [1], M. Fátima M. Piedade [1,2] and Catarina V. Esteves [1,3,*]

[1] Centro de Química Estrutural, Institute of Molecular Sciences, Faculdade de Ciências, Universidade de Lisboa, 1749-016 Lisboa, Portugal

[2] Centro de Química Estrutural, Institute of Molecular Sciences, Instituto Superior Técnico, Universidade de Lisboa, 1049-001 Lisboa, Portugal

[3] Departamento de Engenharia Química e Biológica, Escola Superior de Tecnologia do Barreiro, Instituto Politécnico de Setúbal, Rua Américo da Silva Marinho, 2839-001 Lavradio, Portugal

* Correspondence: caesteves@fc.ul.pt

**Abstract:** Solubility and crystallization studies of a monocarboxylic derivative of pyridine, picolinic acid (2-pyridinecarboxylic acid), were undertaken as a need for new data in the literature was identified. Moreover, comparative studies of structurally related small molecules, such as these pyridinecarboxylic acid isomers (picolinic acid (PA), nicotinic acid (NA, also known as Niacin or vitamin $B_3$), and isonicotinic acid (IA)), can contribute to a larger goal of identifying optimal crystallization conditions. Indeed, vitamin $B_3$ has been thoroughly explored in literature, whilst IA and, particularly, PA have received less attention. Hence, results on both the solubility (obtained through the gravimetric method) and solid-state structure (investigated by means of PXRD) of PA, at different temperatures, in three polar solvents: water, ethanol (both protic solvents) and acetonitrile (aprotic solvent) are presented in this work. These results indicate that PA is very soluble in water (for $T \approx 293$ K, $C_{PA} \approx 862.5$ g·kg$^{-1}$), way less soluble in ethanol ($C_{PA} \approx 57.1$ g·kg$^{-1}$), and even less in acetonitrile ($C_{PA} \approx 17.0$ g·kg$^{-1}$). The crystallization outcome was analyzed in comparison with its family of compounds data, revealing that two polymorphic forms were identifiable for PA, and that no hydrates or solvates were found.

**Keywords:** picolinic acid; solubility; crystallization; polymorphism





## 1. Introduction

Crystallization is an ancient method often used to obtain a crystalline solid from a solution [1]. A crystal can be defined as a highly ordered homogeneous solid that forms a three-dimensional pattern made up of atoms, molecules or ions that repeats itself in the crystal lattice three dimensions [2]. Nevertheless, crystallization can lead to different arrangements of the atoms and/or molecules, a phenomenon known as polymorphism [3]. Polymorphs can have different physical, and chemical properties, such as crystal habit, melting point, color, density, dissolution rate, and solubility [3]. In order to obtain only the desired polymorph, it is essential to study the exact conditions under which it can be synthesized/isolated [4]. The study of polymorphism is necessary for several industries where it plays an active role, for example, in the pharmaceutical industry [5]. Properties such as solubility and dissolution rate are very important in this industry, and the existence of polymorphism can have a big impact on them. Thus, solubility studies and the detection and description of the polymorphism of substances are crucial. Moreover, bioactive organic compounds' solubility has been a topic of significant interest in the latest years [6–8]. To have a desired pharmacological response of a given bioactive compound, the appropriate concentration in systemic circulation must be enhanced and, for that, its solubility should be well known.

Our effort was to contribute to the cumulative study of a family of compounds with systematic variations in their molecular structure. On the one hand, more solubility data was made available and, on the other, such investigations might help to clarify the molecular

mechanisms that take place during crystallization. In this work, we investigate picolinic acid (PA) which has fewer solubility and crystallization data available in the literature. PA and the isomers isonicotinic (IA) and nicotinic (NA) acids, compounds that only differ from hydroxynicotinic (HNA) acids (also studied by us [9]) due to their hydroxy group, constitute an excellent family model for this study (Figure 1).

**Figure 1.** Molecular structures of picolinic, nicotinic, and hydroxynicotinic acids.

There are three single crystal X-ray diffraction (SCXRD) structures of PA available in the literature [10–12] all collected at room temperature (r.t.). They all comprise the monoclinic crystal system with the variation of the space group from $P2_1/a$ to $C2/c$. Thus, two polymorphic forms were identifiable for PA, although no hydrates or solvates were found. PA was also used before in cocrystallization studies [13], in the synthesis of lanthanide-organic polymers [14], and as a chelating moiety of larger compounds for the formation of metal complexes [15–18]. Structures of PA were obtained by SCXRD, both at r.t. and at 150 K, which constitutes a novelty, and the $C2/c$ space group was found. The new solubility data of PA, was obtained through the gravimetric method, and the solid-state structure was acquired by X-ray powder diffraction (PXRD), at different temperatures, in three polar solvents: water, ethanol (both protic solvents) and acetonitrile (aprotic solvent). To the best of our knowledge, it is the first time that all the solids in equilibrium with the solvents under study at different temperatures were fully screened by PXRD. The data reported by us for PA, as well as the data existing in the literature for nicotinic and isonicotinic acids (NA and IA), are now condensed in one place easily accessible for scientists or industrials.

## 2. Materials and Methods

Picolinic acid (PA) was purchased from a commercial source (Alfa Aesar, mass fraction 0.999) and was used without further purification in the solubility studies. For further crystallization investigations it was purified by sublimation at 350.15 K and 1.3 Pa both using a cold finger system (please see Figure S2 for a microscope image of the crystals obtained) and on a Petri dish (see Figure S3). The latter crystals were used in an SCXRD experiment at $150 \pm 2$ K. The compound was characterized in terms of phase purity and chemical purity by PXRD (see Figure 2 for a comparison between this pattern and the ones obtained for the PA as supplied and the one estimated from the most recent structure in the literature [11]), HPLC-ESI/MS (Figure S1, Supplementary Data), by Thermogravimetric analysis (Figure S4), DSC (Figure S5), and NMR (Figure S6). The mass fraction purity of the PA given by high-performance liquid chromatography electrospray mass spectrometry (HPLC-ESI/MS) analysis was 0.99999. $^1$H RMN (400 MHz, D$_2$O): δ 8,599 (1H, *d*, H$_{Pyridine}$), 8,485 (1H, *t*, H$_{Pyridine}$), 8,182 (1H, *d*, H$_{Pyridine}$), 7,959 (1H, *t*, H$_{Pyridine}$) ppm. ESI-MS Calcd. for $[C_6H_5NO_2 + H]^+$: m/z 124.12. Found: m/z 123.9. The powder pattern obtained at $298 \pm 2$ K was indexed as monoclinic (Tables S2 and S3 see Supplementary Materials), space group $C2/c$; $a = 21.2215(1)$ Å, $b = 3.8295(2)$ Å, $c = 13.9497(6)$ Å, $\beta = 108.08(9)°$. These values are in good agreement with those from the literature, found by SCXRD, carried out at r.t. [10]: $C2/c$, $a = 21.262(6)$ Å, $b = 3.837(4)$ Å, $c = 13.972(4)$ Å, and $\beta = 108.02(2)°$, and with those obtained in this work by SCXRD at $150 \pm 2$ K: $a = 21.2110(17)$ Å, $b = 3.7625(3)$ Å, $c = 13.9555(11)$ Å, and $\beta = 107.653(3)°$. Table S1 summarizes relevant information on the provenance and mass fraction purity of the materials used in this work.

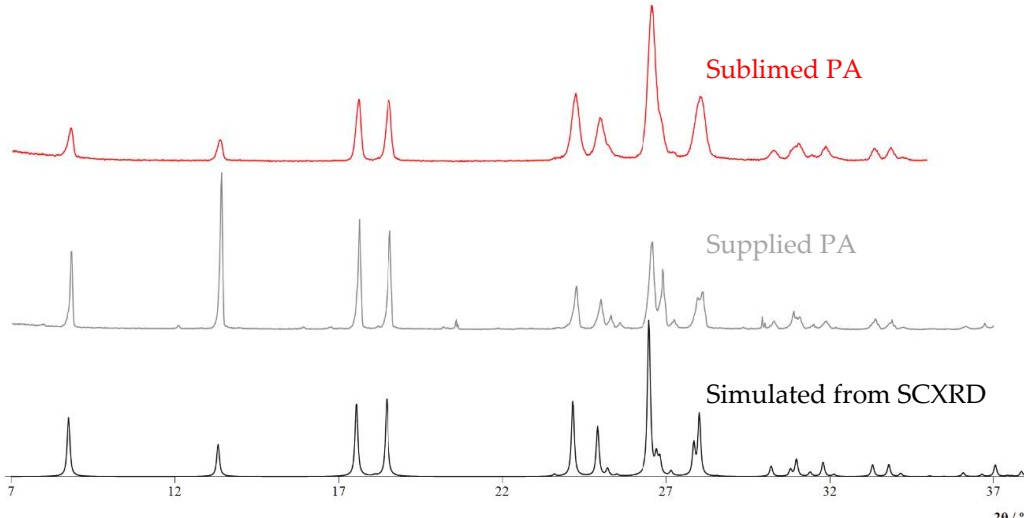

**Figure 2.** Comparison of X-ray diffraction patterns obtained at 293 ± 2 K for: the sublimed PA (red), the PA as supplied (gray), and the pattern estimated from the SCXRD structure found in the literature [11] (black). All the diffractograms were normalized to the peak of highest intensity ($I_n$) and plotted using EasyGraphII [19].

The $^1$H NMR spectrum was acquired on a Bruker Avance II+ 400 ($^1$H at 400.13 MHz) spectrometer at 293 ± 2 K. NMR peak assignments are based on peak integration and multiplicity.

High-performance liquid chromatography electrospray mass spectrometry (HPLC-ESI/MS) analyses were executed on an HPLC Dionex Ultimate 3000 system, connected to an LCQ Fleet ion trap mass spectrometer outfitted with an ESI ion source (Thermo Scientific Portugal). Chromatographic separations were performed on a Phenomenex C18 Luna® column 100 Å (150 × 4.6 mm, 5 µm particle size). The mobile phase was a mixture of 0.1% (v/v) formic acid solution in water (A) and acetonitrile (B). The elution gradient was: 0 min, 50% B; 5 min 100% B; 7–12 min, 50% B. The injected volume was 10 µL, the flow rate 350 µL·min$^{-1}$, and the temperature of the column was kept at 308 K. Mass spectra were attained in the ESI positive and negative modes, under the following conditions: ion spray voltage, ±4.5 kV; capillary voltage, +16 V or −18 V; tube lens offset, +63 V or −125 V; sheath gas ($N_2$), pressure 80 arbitrary units; auxiliary gas ($N_2$) pressure, 20 arbitrary units; capillary temperature, 573 K. Spectra typically corresponded to an average of 20–35 scans and were recorded in the range between 100–800 Da. Data acquisition and processing were executed using the Xcalibur 2.2 software.

PXRD analyses were performed on a Philips X'Pert PRO apparatus fitted with an X'Celerator detector with automatic data acquisition (X'Pert Data Collector, v2.0b, software) and a vertical goniometer PW 3050/60. Copper was used as the source of Kα radiation. The tube current intensity and potential difference were 30 mA and 40 kV, respectively. The diffractograms were recorded at 293 ± 2K, in the range 7 to 35 (°2$\theta$), in the continuous mode with a step size of 0.017 (°2$\theta$), and an accumulation time of 20 s per step. The samples were mounted on a silicon sample holder. The indexation of the powder patterns was performed using the program Chekcell [20]. The assessment of phase purity was made by comparing the X-ray powder diffraction (PXRD) patterns of the PA slurries, recorded at 298 ± 2 K, with the corresponding diffractograms simulated from SCXRD data (please see below). The simulations were completed with the Mercury 2020.2.0 (Build 290188) program [21].

The crystal structure of PA at 150 ± 2 K was solved from single crystal X-ray diffraction data. A small prismatic and colorless crystal, achieved by sublimation, using a petri dish was used. A summary of the crystal data, structure solution, and refinement parameters is in Table 1.

**Table 1.** Crystal data and structure refinement parameters for the PA.

| | |
|---|---|
| CCDC nr | 2209521 |
| Formula | $C_6H_5NO_2$ |
| MW | 123.11 |
| crystal system | Monoclinic |
| space group | *C2/c* |
| wavelength/Å | *0.71073* |
| *T*/K | 150 (2) |
| *a*/Å | 21.2110 (17) |
| *b*/Å | 3.7625 (3) |
| *c*/Å | 13.9555 (11) |
| α/deg | 90 |
| β/deg | 107.653 (3) |
| γ/deg | 90 |
| *v*/Å$^3$ | 1061.29 (15) |
| *F*(000) | 512.0 |
| Z | 8 |
| λ, Å (MoK$_\alpha$) | 0.71073 |
| $D_{calc}$/g cm$^{-3}$ | 1.541 |
| $\mu$/mm$^{-1}$ | 0.118 |
| θ range/deg | 2.015 to 35.905 |
| Limiting indexes | $-34 \leq h \leq 26$ |
| | $-6 \leq k \leq 6$ |
| | $-22 \leq l \leq 22$ |
| $R_{int}$ | 0.0632 |
| reflns collect | 18364 |
| unique reflns | 2478 |
| GOF on $F^2$ | 1.098 |
| $R_{indexes}$ (all data) | $R_1 = 0.0632$ |
| | $wR_2 = 0.1568$ |
| Largest diff peak and hole /eÅ$^{-3}$ | 0.695 and $-0.402$ |

The experiment was carried out on a Bruker AXS-KAPPA APEX II area detector diffractometer. The crystal was coated with Paratone-N oil and mounted on a Kaptan loop. A graphite-monochromated MoK$_\alpha$ ($\lambda$ = 0.71073 Å) radiation source operating at 50 kV and 30 mA was used. For further details please refer to Abhinav et al. [22]. Structural representations were prepared with Mercury 2020.2.0 (Build 290188) [21].

DSC studies were carried out up to 600 K on a Perkin-Elmer DSC 7 apparatus, controlled by a TAC 7/DX thermal analysis unit. It is operated by a computer running Pyris V 7.0 software from Perkin-Elmer. The sample masses used were between 2 and 5 mg and weighted with a precision of $\pm 0.1$ µg on a Mettler XP2U ultra-micro balance. Sealed aluminium crucibles were used, with punctured lids in the case of the hydrate and solvate materials. The experiments were performed under a flow of nitrogen (Praxair 5.0) of 25 cm$^3$min$^{-1}$. The heating rate used was β = 5 K·min$^{-1}$. The temperature scale of the apparatus was calibrated at each heating rate by taking the onset of the fusion peaks of indium (Perkin Elmer; 99.999%; $T_{fus}$ = 429.75 K, $\Delta_{fus}h^\circ$ = 28.45 J·g$^{-1}$), lead (Goodfellow, 99.995%, $T_{fus}$ = 600.61 K), and zinc (Perkin-Elmer, 99.999%, $T_{fus}$ = 692.65 K). The calibration of the heat flow scale was based on the area of the fusion peak of indium.

Thermogravimetric experiments were performed on a PerkinElmer TGA7 apparatus. The balance chamber was kept under a nitrogen flow (Air Liquide N45) of 38 cm$^3$·min$^{-1}$. 2 and 6 mg of the samples were placed in an open platinum crucible. The maximum temperature used was 600 K and each sample was heated at a rate of 5 K·min$^{-1}$. The sample purge gas was nitrogen (Praxair 5.0) at a flow rate of 22.5 cm$^3$·min$^{-1}$. The mass scale of the instrument was calibrated with a standard 100 mg weight and the temperature calibration was based on the measurement of the Curie points (TC) of alumel alloy (PerkinElmer, TC = 427.35 K) and nickel (PerkinElmer, TC = 628.45 K) standard reference materials.

Equilibrium solubility measurements in the 293.15 K to 323.15 K range were carried out by the gravimetric method [23]. The apparatus and procedure were essentially the same as previously described [24]. In brief, a suspension of picolinic acid was placed in ~100 cm$^3$ of each solvent and magnetically stirred for one week, under a nitrogen atmosphere, inside glass reactors consisting of a Schlenk tube with an external jacket for circulating water from a thermostatic bath. Two polar protic solvents were used for the measurements (H$_2$O and EtOH) and one polar aprotic (MeCN). The aqueous solubility studies were carried out in distilled and deionized H$_2$O from a Milli-Q Plus system (conductivity 0.1 µS·cm$^{-1}$). The solubility studies in EtOH were performed using Carlo Erba Reagents Ethanol absolute ($v/v$): 99.9%, and in MeCN were performed with Chem-Lab ($v/v$): 99.9%. The bath temperature was controlled to ±0.01 K by a Thermomix UB B-Braun unit and a HAAKE K10 immersion cooler. The temperature of the suspension was monitored with a resolution of ±0.01 K by a Labfacility ceramic encapsulated Pt100 sensor. The sensor was inserted in a glass tube containing Baysilone M350 oil to improve thermal contact and was connected in a four-wire configuration to an Agilent HP34901A 20 channel multiplexer adapted to a $6\frac{1}{2}$ digits Agilent HP34970A multimeter. This sensor had been calibrated against a reference platinum resistance thermometer, calibrated at an accredited facility in accordance with the International Temperature Scale ITS-90. At the end of the equilibration period stirring was stopped and three samples of the saturated solution, each of ~2 cm$^3$, were extracted using a pre-thermostatized syringe adapted to a µfilter (VWR syringe filters with a diameter of 25 mm, and 0.2 µm porous, cellulose acetate membrane—for aqueous samples—and PTFE membrane for the EtOH and MeCN solutions) and a Hamilton 7748-06 stainless steel needle. The aliquots were transferred to a previously weighed glass vial, which was weighted a second time when loaded with the solution and a third time after the solution was taken to dryness. The weightings were performed with a precision of ±0.01 mg on a Mettler Toledo XS205 balance. The mole fraction of the picolinic acid in the saturated solutions was calculated from the following Equation (1):

$$x_{\mathrm{PA}} = \frac{M_{\mathrm{solvent}}(m_{\mathrm{vial+residue}} - m_{\mathrm{vial}})}{M_{\mathrm{solvent}}(m_{\mathrm{vial+residue}} - m_{\mathrm{vial}}) + M_{\mathrm{PA}}(m_{\mathrm{vial+solution}} - m_{\mathrm{vial+residue}})} \tag{1}$$

The measurements were carried out both in ascending and descending temperature modes. The one-week equilibration time was deduced from a preliminary experiment carried out at $T$ = 298 K, where the concentration of the PA was found to be stable after such a time interval.

The pH measurements were made at 293.2 K, with a TIM900 pH meter, fitted with an InLab Routine pH electrode from Mettler Toledo.

## 3. Results and Discussion

### 3.1. Solubility Determinations

This work's results regarding the solubility determinations on PA in mole fractions, $x_{\mathrm{PA}}$, are presented in Table 2, for the two polar protic solvents, H$_2$O and EtOH, and the one polar aprotic solvent, MeCN, used for the measurements.

**Table 2.** Temperature dependency of the mole fraction ($x_{\mathrm{PA}}$) equilibrium solubilities of PA.

| H$_2$O | | EtOH | | MeCN | |
|---|---|---|---|---|---|
| $T$/K | $x_{\mathrm{PA}} \cdot 10$ | $T$/K | $x_{\mathrm{PA}} \cdot 10^2$ | $T$/K | $x_{\mathrm{PA}} \cdot 10^3$ |
| 293.62 | 1.1009 ± 0.0318 | 293.65 | 2.0923 ± 0.0140 | 293.65 | 5.7675 ± 0.6945 |
| 298.59 | 1.1807 ± 0.0053 | 298.60 | 2.2951 ± 0.0065 | 298.54 | 7.5130 ± 0.1489 |
| 303.53 | 1.2877 ± 0.0038 | 303.48 | 2.9054 ± 0.0065 | 303.43 | 7.6812 ± 0.2602 |
| 308.59 | 1.3426 ± 0.0102 | 308.46 | 3.4747 ± 0.0196 | 308.52 | 11.616 ± 0.096 |
| 313.42 | 1.4280 ± 0.0138 | 313.42 | 3.8851 ± 0.0354 | 313.42 | 13.991 ± 0.312 |
| 318.55 | 1.4003 ± 0.0013 | 318.41 | 4.9302 ± 0.0191 | 318.25 | 15.711 ± 0.393 |
| 323.38 | 1.4953 ± 0.0625 | 323.28 | 5.8491 ± 0.0223 | 323.22 | 22.024 ± 0.150 |

The $x_{PA}$ versus $T$ data in Table 2 was fitted by the least squares regression to the Equation (2):

$$\ln x_{PA} = \frac{m}{T} + b \tag{2}$$

The values obtained for the $m$ and $b$ parameters are presented in Table 3; as well as the determination coefficients ($R^2$); for 95% probability

**Table 3.** Parameters for equation (2) and coefficients ($R^2$) in the calculation of $\ln x_{PA}$.

| Solvent | $-m$ | $b$ | $R^2$ | $\sigma_{\ln x_{PA}}$ |
|---------|------|-----|-------|-----------------------|
| H$_2$O | 1001.3 ± 101.7 | 1.2251 ± 0.3319 | 0.96038 | 3.1348 |
| EtOH | 3360.2 ± 151.7 | 7.5277 ± 0.4926 | 0.98991 | 4.1793 |
| MeCN | 4070.9 ± 197.8 | 8.7215 ± 0.6408 | 0.99064 | 79.282 |

The $x_{PA}$ versus $T$ data were plotted in Figure 3. The analysis of such data reveals that PA is very soluble in water (for $T \approx 293$ K, C$_{PA} \approx 862.5$ g·kg$^{-1}$), way less soluble in ethanol (C$_{PA} \approx 57.1$ g·kg$^{-1}$) and even less in acetonitrile (C$_{PA} \approx 17.0$ g·kg$^{-1}$).

In order to better understand the solubility of PA a comparison with nicotinic acid (NA) was drafted and, for that, a search for the solubility data of the latter was made. In the literature, NA solubility values for water (H$_2$O), ethanol (EtOH), and acetonitrile (MeCN) are available, as well as for acetone (AcO), diethyl ether (Et$_2$O), dimethyl sulfoxide (DMSO) [24], and in four additional alcoholic solvents: *n*-butanol (*n*-BuOH), 1-pentanol, 1-hexanol and 2-butanol [25]. There is no evidence of NA solvate development in H$_2$O, or in any of the reported organic solvents. The parameters of Equation (2) in the literature for NA [24], in H$_2$O (obtained between 283 K and 333 K) are as follows: $b = (2.04994 \pm 0.01833)$, $m = (-2394.68 \pm 55.67)$ ($R^2 = 0.9978$), in EtOH: $b = (5.04873 \pm 0.27795)$, $m = (-3172.75 \pm 85.22)$ ($R^2 = 0.9971$) and, in MeCN: $b = (8.52352 \pm 0.58667)$, $m = (-5075.54 \pm 178.63)$ ($R^2 = 0.9902$). These solubility results and the ones here presented, showed, the following trends (~293 K) in H$_2$O: PA >>> NA, in EtOH: PA >> NA and, in MeCN: PA > NA.

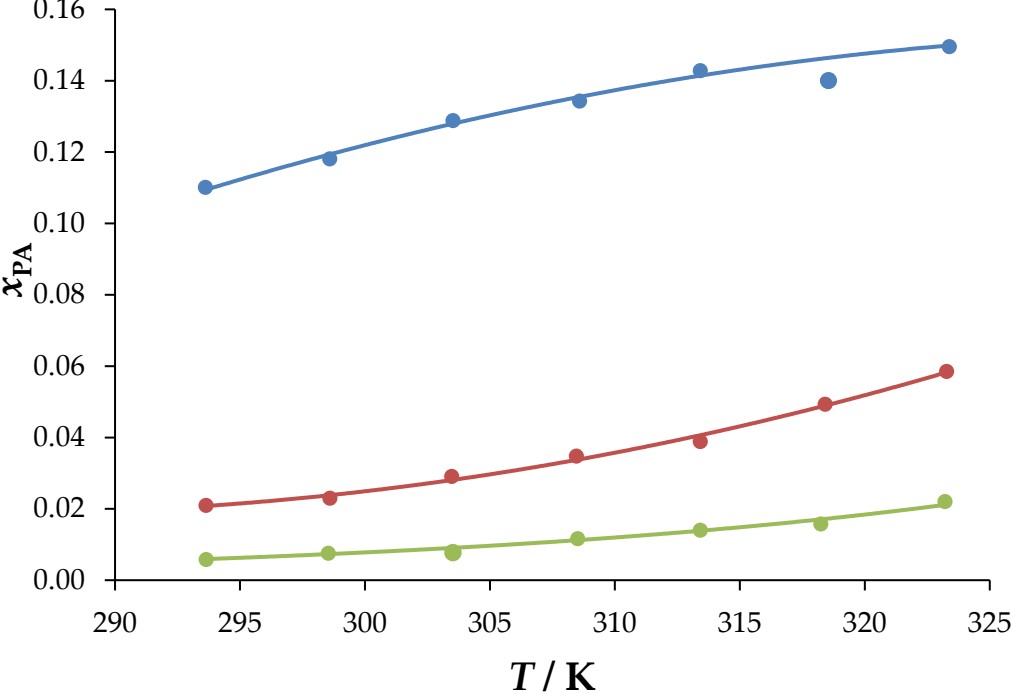

**Figure 3.** $x_{PA}$ versus $T$ plot of picolinic acid in: H$_2$O (blue), EtOH (red), and MeCN (green).

Further literature scouting was made for isonicotinic acid (IA), so that a comparison could be made also with this isomer. For IA only data in $H_2O$ and EtOH was found [26]. The solubility trend, drawn from the reported values of IA and our values of PA, is similar to the one observed for NA. Indeed, PA is the most soluble isomer in all compared solvents, followed by NA, and IA, which is the less soluble isomer. The following trends (at ~293 K) were found in $H_2O$: PA >>> NA > IA, and in EtOH: PA >> NA > IA. As different concentration units were reported in the literature, Table S4 was constructed to summarize all the information, and can be found in the Supplementary Materials.

To try to explain the solubility trend observed for PA ($H_2O$ >>> EtOH >> MeCN); an analysis of the correlations between the $\ln x_{PA}$; calculated from Equation (2) and several parameters that represent various properties of the solvents used were made. The parameters were calculated as described before in the literature [24] and were the following: dipole moment ($\mu$); the Hansen polar solubility parameter ($\delta_P$); the normalized Dimroth-Reichardt polarity parameter ($E_T^N$); the Kosower polarity parameter ($Z$); the solubility dispersion parameter ($\delta_D$); the Hildebrand solubility parameter ($\delta$); the molar refractivity ($MR$); the Kamlet-Taft combined dipolarity/polarizability parameter ($\pi^*$), the Kamlet-Taft parameters of donor ($\alpha$) and acceptor ($\beta$) of hydrogen bonds and the Hansen hydrogen bond solubility parameter ($\delta_H$). The parameters for the solvents water; ethanol; and acetonitrile at a temperature of 298 K are shown in Table 4

**Table 4.** Values of the parameters used ($T$ = 298 K) to describe the three solvents (water, ethanol, and acetonitrile).

| Parameter | Water | Ethanol | Acetonitrile |
|---|---|---|---|
| $\mu \cdot 10^3 / C \cdot m$ | 6.2 | 5.5 | 13 |
| $\delta / J^{1/2} \cdot cm^{-3/2}$ | 47.9 | 26 | 24.1 |
| $\delta_D / MPa^{1/2}$ | 15.5 | 15.8 | 15.3 |
| $\delta_P / MPa^{1/2}$ | 16 | 8.8 | 18 |
| $\delta_H / MPa^{1/2}$ | 42.3 | 19.4 | 6.1 |
| $\alpha$ | 1.17 | 0.86 | 0.19 |
| $\beta$ | 0.47 | 0.75 | 0.4 |
| $\pi^*$ | 1.09 | 0.54 | 0.66 |
| $Z / kcal \cdot mol^{-1}$ | 94.6 | 79.6 | 71.3 |
| $E_T^N / kcal \cdot mol^{-1}$ | 1.000 | 0.654 | 0.460 |
| $MR / cm^3 \cdot mol^{-1}$ | 3.71 | 12.936 | 11.118 |

Linear regressions were performed, using the least squares method, of $\ln x_{PA}$ and each parameter. The slope ($m$), y intercept ($b$), and correlation coefficient ($R^2$) are presented in Table 5.

**Table 5.** Slope ($m$), y intercept ($b$), and correlation coefficient ($R^2$) of the parameters considered.

| Parameter | $m$ | $b$ | $R^2$ |
|---|---|---|---|
| $\mu \cdot 10^3 / C \cdot m$ | −0.2728 | −1.501 | 0.5788 |
| $\delta / J^{1/2} \cdot cm^{-3/2}$ | 0.1053 | −7.188 | 0.8800 |
| $\delta_D / MPa^{1/2}$ | 1.804 | −31.76 | 0.09341 |
| $\delta_P / MPa^{1/2}$ | −0.03374 | −3.266 | 0.01208 |
| $\delta_H / MPa^{1/2}$ | 0.08099 | −5.577 | 0.9971 |
| $\alpha$ | 2.826 | −5.838 | 0.9087 |
| $\beta$ | 0.7375 | −4.145 | 0.008457 |
| $\pi^*$ | 4.136 | −6.904 | 0.6486 |
| $Z / kcal \cdot mol^{-1}$ | 0.1255 | −14.02 | 0.9956 |
| $E_T^N / kcal \cdot mol^{-1}$ | 5.419 | −7.565 | 0.9960 |
| $MR / cm^3 \cdot mol^{-1}$ | −0.2486 | −1.446 | 0.6693 |

The main solvent parameters that are related to the PA solubility are the Hildebrand solubility ($\delta$), the Hansen Hydrogen bond solubility ($\delta_H$), the Kamlet-Taft bond-donor solu-

bility of Hydrogen ($\alpha$), the Kosower polarity parameter ($Z$) and the normalized Dimroth-Reichardt polarity parameter ($E_T^N$). The parameter $\delta$ indicates the work required to separate the solvent molecules, to create a cavity large enough to accommodate the solvent and, generally, solubility is greater the closer the value is to the value of $\delta$ solute, so the value for the PA should be quite close to that of water (47.9 $J^{1/2} \cdot cm^{-3/2}$). This factor, $\delta$, considers the contribution of three of the parameters $\delta_D$ (contribution of non-polar/dispersive forces), $\delta_P$ (contribution of permanent dipole–permanent dipole molecular forces) and $\delta_H$ (contribution of hydrogen bond forces). As can be seen from Table 5, $\delta_H$ has a very high value of $R^2$, compared to $\delta_P$ and $\delta_D$. This suggests that PA is much more soluble in solvents capable of making strong Hydrogen bonds with each other, as they can make stronger Hydrogen bonds with PA. This will surely influence the aggregation behavior of PA making it less prone to aggregate in such solvents. Furthermore, the value of $R^2$ in relation to the $\alpha$ parameter also indicates that PA is more soluble in compounds that are good Hydrogen bond donors. This is indicative that PA must behave as a Hydrogen bond acceptor. Indeed, the ability of the PA o-carboxylic acid group to form five-membered intramolecular O–H . . . N rings has raised some interest [13]. The parameters $Z$ and $E_T^N$ indicate the polarity of the solvent, which will be higher the value associated with it. Thus, PA is more soluble the more polar the solvent, as would be expected.

Picolinic Acid Solid Forms in Equilibrium with the Saturated Aqueous, Ethanol and Acetonitrile Solutions

In order to gain knowledge about the most stable solid forms at a given temperature, the solids in equilibrium with the saturated solutions of PA, in different solvents, as a function of temperature were studied. In Table 6, a summary of the known solid forms of the PA, used for comparison, is presented. Also, for discussion and comparison reasons the known forms for NA and IA are also presented.

**Table 6.** Picolinic, nicotinic and isonicotinic acids crystal systems, spaced groups and cell parameters gathered from literature SCXRD data.

| Compound | Crystal System | Space Group | $a$/Å | $b$/Å | $c$/Å | $\alpha$/° | $\beta$/° | $\gamma$/° | Reference |
|---|---|---|---|---|---|---|---|---|---|
| PA | Monoclinic | P2$_1$/a | 13.97 | 3.84 | 10.62 | 90 | 107.9 | 90 | [12] |
| | | | 21.267 | 3.831 | 13.970 | 90 | 108.01 | 90 | [10] |
| | | C2/c | 21.262 (6) | 3.837 (4) | 13.972 (4) | 90 | 108.02 (2) | 90 | [11] |
| | | | 21.211 (17) | 3.7625 (3) | 13.9555 (11) | 90 | 107.653 (3) | 90 | This work |
| NA | Monoclinic | P2$_1$/c | 7.175 (2) | 11.682 (2) | 7.220 (2) | 90 | 113.38 (5) | 90 | [27] |
| | | | 7.162 | 11.703 | 7.242 | 90 | 113.2 | 90 | [28] |
| | | | 7.186 (2) | 11.688 (3) | 7.231 (2) | 90 | 113.55 (6) | 90 | [29] |
| | | | 7.303 (11) | 11.693 (2) | 7.33 (3) | 90 | 113.68 (14) | 90 | [30] |
| | | | 7.41 (3) | 11.692 (2) | 7.377 (11) | 90 | 114.45 (14) | 90 | [30] |
| | | | 7.1672 (5) | 11.6710 (6) | 7.1057 (6) | 90 | 114.785 (10) | 90 | [31] |
| IA | Triclinic | P-1 | 7.231 (1) | 7.469 (1) | 6.392 (1) | 114.88 (2) | 106.19 (1) | x103.66 (2) | [32] |

The PA structure obtained in this work (Figure 4) is in good agreement with the structure published by Hamazaki et al. [11]. The crystal structure reported by Tamura et al. [12] could not be reproduced, despite our several attempts. However, taking into account this literature report there are two polymorphic forms identifiable for PA. It is noteworthy that no hydrates or solvates were found, neither for PA, nor NA and IA. In the case of NA only one form was found, as was the case of IA. Overall, ten structures are reported in the literature for PA (three), NA (six) and IA (one) family of compounds.

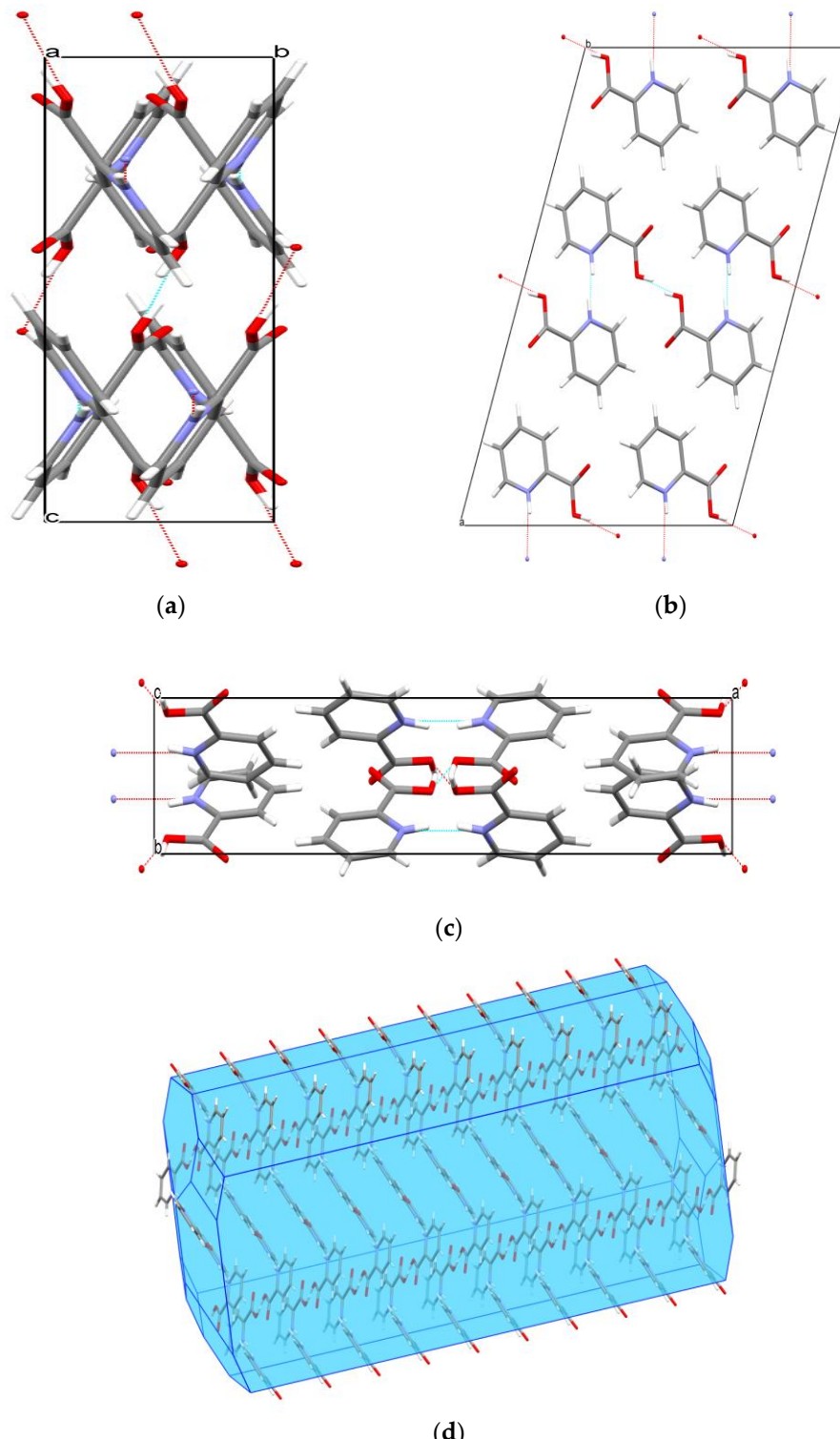

(**a**)

(**b**)

(**c**)

(**d**)

**Figure 4.** Views of the crystal (small, prismatic, and colorless, achieved by sublimation on a petri dish) structure of picolinic acid (PA) along a (**a**), b (**b**), c (**c**) axes, and the Bravais-Friedel-Donnay-Harker (BFDH) morphology (**d**) determined from single crystal X-ray diffraction data using Mercury 2020.2.0 (Build 290188) program [21].

All the solid forms in equilibrium with the saturated PA solutions, during the solubility experiments, were analyzed by PXRD (Tables S5–S27), to verify if the crystal structure remains unaltered over the temperature range studied. For the aqueous solutions (see Figure 5)

the diffractogram of the solid PA in equilibrium with its saturated solution at 308 K might strike out as different from the others, with a decrease in the intensity of many of the stripes. It is likely that this diffractogram presents such intensity decrease in some stripes due to the sample taking preferential orientations during the preparation and acquisition of its PXRD. For the remaining diffractograms, there are some differences noted: for $2\theta \approx 13°$ (blue rectangles), for the solids filtered off the saturated PA solutions at 293 to 303 K, the stripes are sharper and less intense, while from 313 to 323 K the stripes are less sharp, more intense and have a "shoulder", that is, they are two barely separated stripes that overlap each other; for $2\theta \approx 26°$ to $\approx 27°$ (green rectangles) at 293 and 298 K there is a stripe that has one shoulder to the left, at 303 K, it is no longer a shoulder and becomes two stripes with some overlap, on the other hand, from 313 to 323 K the shoulder is the left stripe and the right stripe appears to have several different stripes that end up being slightly overlapping; for $2\theta \approx 30°$ to $\approx 32°$ (brown rectangles), at 293 and 298 K, there are several lines, easily differentiated, whereas from 313 K to 323 K the lines are overlapping. Besides these observations, none of them constitute sufficient evidence for the appearance of a new polymorph.

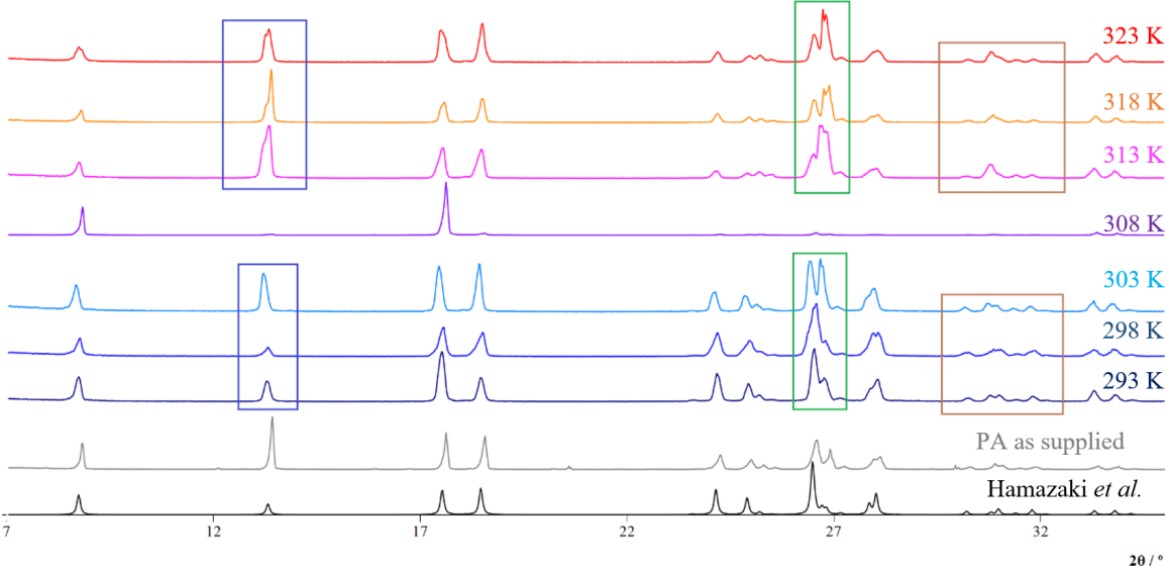

**Figure 5.** Comparison of X-ray diffraction patterns obtained at 293 $\pm$ 2 K for solid forms in equilibrium with the saturated PA aqueous solutions stabilized for one week at different temperatures. The PA as supplied (gray), and the pattern estimated from the SCXRD structure found in the literature [11] (black), are also plotted. All the diffractograms were normalized to the peak of highest intensity ($I_n$) and plotted using EasyGraphII [19].

In the case of the PA solids in equilibrium with its saturated ethanol solutions (Figure 6), the diffractograms are quite similar to each other. Still, there is a small difference for the $2\theta$ values between $\approx 26°$ and $\approx 27°$ (green rectangle). For 293 and 303 K there is a stripe that has one shoulder to the right, at 303 K the stripe is broader and the shoulder less noticeable, and from 308 up to 323 K the shoulder has a different profile and is most often further apart from the main stripe. Nonetheless, and once again, in our opinion, these evidences are insufficient to report a new PA polymorph.

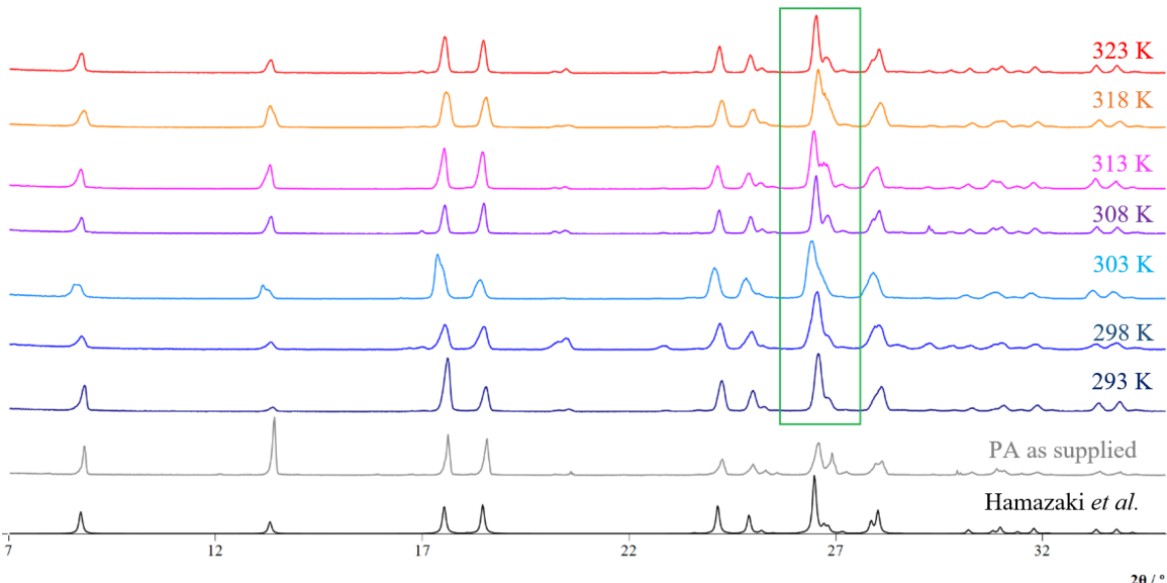

**Figure 6.** Comparison of X-ray diffraction patterns obtained at $293 \pm 2$ K for solid forms in equilibrium with the saturated PA ethanol solutions stabilized for one week at different temperatures. The PA as supplied (gray), and the pattern estimated from the SCXRD structure found in the literature [11] (black), are also plotted. All the diffractograms were normalized to the peak of highest intensity ($I_n$) and plotted using EasyGraphII [19].

For PA filtered off its saturated acetonitrile solutions (Figure 7) the diffractograms are very similar to each other with no significant differences in this case.

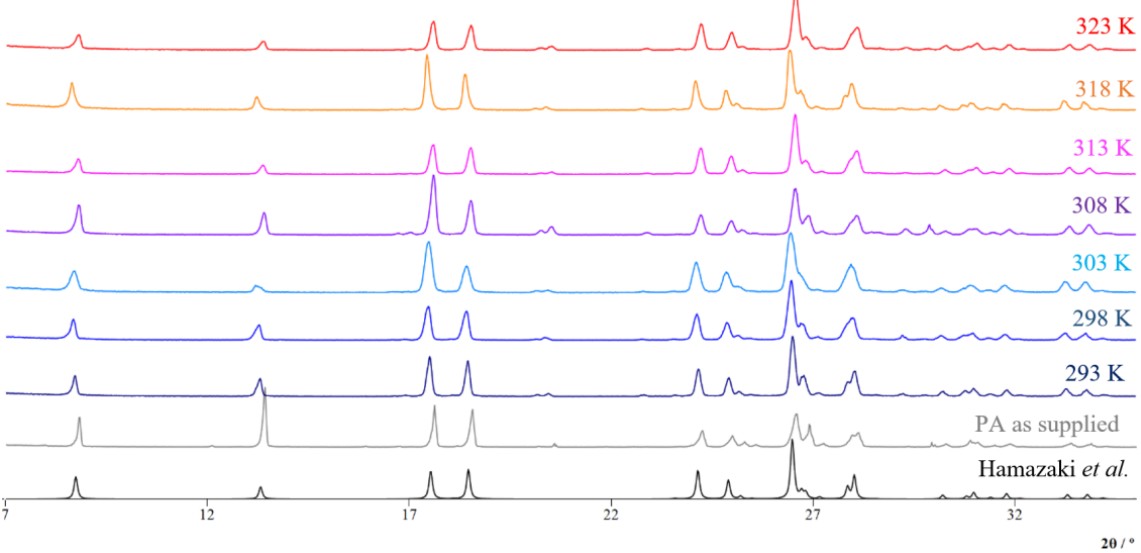

**Figure 7.** Comparison of X-ray diffraction patterns obtained at $293 \pm 2$ K for solid forms in equilibrium with the saturated PA acetonitrile solutions stabilized for one week at different temperatures. The PA as supplied (gray), and the pattern estimated from the SCXRD structure found in the literature [11] (black), are also plotted. All the diffractograms were normalized to the peak of highest intensity ($I_n$) and plotted using EasyGraphII [19].

### 3.2. Acid–Base Behavior of Picolinic, Nicotinic and Isonicotinic Acids, Speciation in Water, and Its Influence on Their Solid-State Structures

The pH value of saturated aqueous solutions of PA, was measured at $298 \pm 2$ K, and found to be about 4 (the corresponding pH value for NA was found to be 3.5 [24]). To know which species might be present at such pH a species distribution diagram was calculated for PA (Figure 8a) using the literature overall ($\beta_i^H$) and stepwise ($K_i^H$) protonation constants found in the literature (Table 7). For comparison purposes, NA and IA values and correspondent speciations were also considered. Protonation constants are fundamental to rationalizing the diverse behavior of each compound in an aqueous solution. For PA, NA, and IA only two protonation constants were expected. Indeed, two low protonation constants are reported ranging from 1 to 2 for the Nitrogen ring and from 4 to 5 for the carboxyl group, in log $K$. Some small variations in the referred values were found in different reports, which are due to slightly diverse working temperatures ($T$) and ionic strengths ($I$).

**Table 7.** Collection of literature overall ($\beta_i^H$) and stepwise ($K_i^H$) protonation constants of PA, NA, and IA in aqueous solution.

| Equilibrium Reaction * | PA | NA | IA |
|---|---|---|---|
| **log $\beta_i^H$** | | | |
| $A + H^+ \rightleftarrows HA$ | -, - | -, - | -, - |
| $A + 2\,H^+ \rightleftarrows H_2A$ | 5.32 [33] [a], 5.40 [34] [b], 5.32 [35] [c] | 4.81 [35] [c], 4.75 [a] | 4.95 [34] [b], 4.86 [35] [c], 4.78 [d] |
| $A + 3\,H^+ \rightleftarrows H_3A$ | 6.40, 7.00, 6.33 | 6.88, 6.81 | 6.65, 6.70, 6.60 |
| **log $K_i^H$** | | | |
| $A + H^+ \rightleftarrows HA$ | -, - | -, - | -, - |
| $HA + H^+ \rightleftarrows H_2A$ | 5.32, 5.40, 5.32 | 4.81, 4.75 | 4.95, 4.86, 4.78 |
| $H_2A + H^+ \rightleftarrows H_3A$ | 1.08, 1.60, 1.01 | 2.07, 2.09 | 1.70, 1.84, 1.82 |

* A denotes the acid of interest; charges were omitted. [a] $T = 298.2$ K, $I = 0.03$ g·L$^{-1}$ NaOAc/HCl solutions. [b] $T = 293 \pm 2$ K, $I = 0.01$ mol·L$^{-1}$ NaOAc solutions, spectroscopic measurements. [c] $T = 295.2$ K, spectroscopic measurements. [d] $T = 298.2$ K, $I = 0.015$ g·L$^{-1}$ NaOAc/AcOH buffer solutions.

A near-ultraviolet absorption study of the pyridine monocarboxylic acids in water and in ethanol [36] has interpreted those changes in the spectra due to solvent being altered from ethanol to water, are consistent with these acids existing primarily in the neutral, undissociated form, in ethanol, and in the anion and zwitterion forms, in water. Such interpretation correlates well with the higher solubility values found for PA and its parent compounds in water. At the pH value determined in this work, PA must be either neutral or zwitterionic, with an abundance near 100% (Figure 8a). Indeed, this relates well with the diffraction patterns, obtained at $293 \pm 2$ K, for the solid forms in equilibrium with the saturated PA aqueous solutions. For NA and IA, the speciations are very similar, and so is the form in which they must be. Only the % of abundance varies and, in this case, at pH 4 NA is the one with a slightly lower one. Moreover, PA is the one with the largest pH interval for the abundance of its AH species, which correlates well with its increased solubility in water. With the analysis of the acid–base behavior of picolinic acid and its speciation in water, not only a reason for its increased solubility in this solvent could be uttered, but also the relation it has on its solid-state structure expressed.

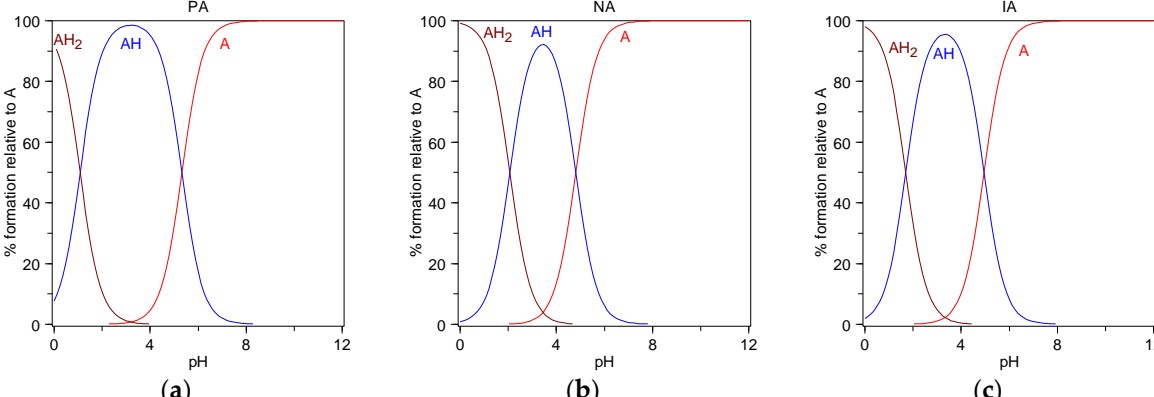

**Figure 8.** Species distribution diagrams for PA (**a**), NA (**b**), and IA (**c**), in aqueous solution, calculated with the protonation constants presented in Table 7.

## 4. Conclusions

This work was performed to contribute to the knowledge gathered regarding the solubility and crystallization of picolinic acid (PA), increasing the available information about this family of compounds. As the temperature and solvent used can affect crystallization and polymorphism emergence, PA was studied at different temperatures in three polar solvents: water, ethanol (both protic solvents), and acetonitrile (aprotic solvent). Its solubility was attained using the gravimetric method and it was found that PA is much more soluble in water than in ethanol or acetonitrile. PA is the most soluble compound by a large difference when compared with its related compounds, NA and IA. This stroked us as a curious observation and might be related to the pH window at which PA finds itself in a zwitterionic or neutral form when in an aqueous solution. Two polymorphic forms were identifiable for PA. It is noteworthy that no hydrates or solvates were found. With this study of PA, the family of PA, NA, and IA, with systematic variations in the molecular structure was attained. Knowing how PA differs from its family of related compounds is important to researchers both in academia and in industry. As in order to synthesize only the desired solid, of a given compound, it is crucial to know the exact conditions at stake or to rely on preexisting literature data. Contributing to this body of work through the study of families of compounds, with systematic variations in their molecular structure, could enhance the prediction of the crystallization outcome.

**Supplementary Materials:** The supporting information can be downloaded at: https://www.mdpi.com/article/10.3390/cryst13030392/s1, Figure S1: UV-Vis and ESI mass spectra of PA; Figure S2: Microscope image of PA crystals obtained through sublimation using a cold finger; Figure S3: Microscope image of PA crystals obtained through sublimation on a Petri dish used for SCXRD; Figure S4: Thermogram of PA obtained through the thermogravimetric analysis (TGA); Figure S5: Thermogram obtained through differential scanning calorimetry (DSC) in the temperature range of 298–423 K of PA; Figure S6: $^1$H-NMR spectrum of a solution of PA in $D_2O$ with a concentration of 126.28 g·kg$^{-1}$; Table S1: Provenance and mass fraction purity of the materials used in this work; Table S2: Indexation of the X-ray Powder Diffraction Pattern of PA, as supplied; Table S3: Indexation of the X-ray Powder Diffraction Pattern of PA, obtained through sublimation using a cold finger; Table S4: Solubility of the picolinic, nicotinic and isonicotinic acid isomers in water, ethanol and acetonitrile at 25 and 30 °C in mol·dm$^{-3}$; Table S5: Indexation of the X-ray Powder Diffraction Pattern of PA, in $H_2O$ at 293.15 K; Table S6: Indexation of PA, in EtOH at 293.15 K; Table S7: Indexation of PA, in MeCN at 293.15 K; Table S8: Indexation of PA, in $H_2O$ at 298.15 K; Table S9: Indexation of PA, in EtOH at 298.15 K; Table S10: Indexation of PA, in MeCN at 298.15 K; Table S11: Indexation of PA, in $H_2O$ at 303.15 K; Table S12: Indexation of PA, in EtOH at 303.15 K; Table S13: Indexation of PA, in MeCN at 303.15 K; Table S14: Indexation of PA, in $H_2O$ at 308.15 K; Table S15: Indexation of PA, in EtOH at 308.15 K; Table S16: Indexation of PA, in MeCN at 308.15 K; Table S17: Indexation of PA, in $H_2O$ at 313.15 K; Table S18: Indexation of PA, in EtOH at 313.15 K; Table S19: Indexation of PA, in MeCN at 313.15 K; Table S20: Indexation of PA, in toluene at

313.15 K; Table S21: Indexation of PA, in cyclohexane at 313.15 K; Table S22: Indexation of PA, in $H_2O$ at 318.15 K; Table S23: Indexation of PA, in EtOH at 318.15 K; Table S24: Indexation of PA, in MeCN at 318.15 K; Table S25: Indexation of PA, in $H_2O$ at 323.15 K; Table S26: Indexation of PA, in EtOH at 323.15 K; Table S27: Indexation of PA, in MeCN at 323.15 K.

**Author Contributions:** Conceptualization, C.V.E. and M.F.M.P.; methodology, C.V.E. and D.S.B.; investigation, C.V.E. and D.S.B.; resources, M.F.M.P.; data curation, C.V.E. and D.S.B.; writing—original draft preparation, C.V.E.; writing—review and editing, C.V.E.; supervision, C.V.E.; project administration, C.V.E. and M.F.M.P.; funding acquisition, M.F.M.P. All authors have read and agreed to the published version of the manuscript.

**Funding:** This research was funded by Fundação para a Ciência e a Tecnologia (FCT), Portugal (projects PTDC/QUIOUT/28401/2017, LISBOA-01-0145-FEDER-028401, UIDB/00100/2020, UIDP/00100/2020 and LA/P/0056/2020).

**Acknowledgments:** The authors thank: M. C. Oliveira (CQE-IST, Portugal) for the HPLC-ESI/MS analyses; D. Silva (CQE-IST, Portugal) and A. Mourato (CQE-FCUL, Portugal) for their help with PXRD; C. E. S. Bernardes, R. G. Simões, A. O. L. Évora, C. S. D. Lopes, I. O. Feliciano and M. E. Minas da Piedade for being always available to helpful discussions and assistance at the Molecular Energetics Lab.

**Conflicts of Interest:** The authors declare no conflict of interest. The funders had no role in the design of the study; in the collection, analyses, or interpretation of data; in the writing of the manuscript; or in the decision to publish the results.

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
