# Peer review of "Solubility and Crystallization Studies of Picolinic Acid"

_crystals, doi:10.3390/cryst13030392_

Round 1

Reviewer 1 Report

This article is devoted to the solubility and crystallization of picolinic acid. The article deals with solubility and crystallization in solvents of different nature (H2O, EtOH, MeCN). The article provides clear calculations and they are sufficiently described. I would recommend making the following fixes:

1. Why did the authors choose this particular object of study? it's pretty simple.

2. The presence in the article of references 50 years ago, of course, indicates a long-standing problem and the ability of the authors to work with old sources of literature. However, I recommend adding more up-to-date links.

3. It is desirable to add more comparison with literary sources. This will give more validity to the conclusions of the authors.

4. Please cite: 10.1016/j.jcrysgro.2012.01.059, 10.1039/C7CE02009K. It is also possible to compare the results of the authors with these literary sources.

5. It is desirable to expand the description of the obtained data.

6. You can quote: 10.1016/j.molstruc.2022.133394.

7. Technical:

- unify all drawings.

- the list of references should be made according to the template.

- It is advisable to double-check the text for typos and errors.

Author Response

Dear Reviewer 1:

“This article is devoted to the solubility and crystallization of picolinic acid. The article deals with solubility and crystallization in solvents of different nature (H2O, EtOH, MeCN). The article provides clear calculations and they are sufficiently described. I would recommend making the following fixes:

  1. Why did the authors choose this particular object of study? it's pretty simple.

R: We chose this object of study due to the lack of understanding, of the events that take place at molecular level, that still permeates the field of crystallization. In our opinion, with simple studies of small, structurally related, molecules, it will be possible to improve crystallization processes related knowledge. For that, we have selected nicotinic acid and its isomers, as a family of related molecules. Nicotinic acid has been thoroughly studied in the literature and in our lab, and one can find data also for isonicotinic acid, however, picolinic acid is less explored. Thus, we have thought about contributing with new data, and with that, try to understand the differences that occur, at solubility and crystallization levels, for such related molecules. With these studies we would like to help establishing a benchmark for crystallization processes, which could be fairly relevant for industry, particularly for the pharmaceutical one.

  1. The presence in the article of references 50 years ago, of course, indicates a long-standing problem and the ability of the authors to work with old sources of literature. However, I recommend adding more up-to-date links.

R: Dear Reviewer, please find more up-to-date literature in the revised version of the manuscript.

  1. It is desirable to add more comparison with literary sources. This will give more validity to the conclusions of the authors.

R: It’s not easy to add more comparison with literature sources in the revised version of the manuscript, as picolinic acid is scarcely studied.

  1. Please cite: 10.1016/j.jcrysgro.2012.01.059, 10.1039/C7CE02009K. It is also possible to compare the results of the authors with these literary sources.

R: Please find the suggested citations in the revised version of the manuscript. However, these articles studies are quite different from ours, as the first one is centred picolinic acid cocrystallisation with dicarboxylic acids, and the second focus on a picolinic acid derivative (5-(3,4-Dicarboxylphenyl)picolinic acid) used for new coordination compounds.  

  1. It is desirable to expand the description of the obtained data.

R: An expansion of the description of the obtained data may be found in the revised version of the manuscript.

  1. You can quote: 10.1016/j.molstruc.2022.133394.

R: I’m sorry but the suggested research article seems a bit far from the scope of this article.

  1. Technical:

- unify all drawings.

- the list of references should be made according to the template.

- It is advisable to double-check the text for typos and errors.”

R: Dear Reviewer, thank you for your suggestions.

Reviewer 2 Report

The manuscript by Esteves et al. reports the solubility of picolinic acid in water, ethanol, and acetonitrile in the temperature range from 293.6 K to 323.4 K. These data are reported in Table 2 and Table 3.

The rest of the manuscript is a retelling of literature data, a series of muzzily obtained correlations, the main conclusion of which is the very common statement that " picolinic acid is more soluble the more polar the solvent", and powder diffraction experiments, which did not give any result. The structure is not very good, the research topic seems  not substantiated and has nothing to do with crystals.

The list of shortcomings:

The aim of the abstract is not to present the study but to summarize its most important results.

Figure 3 and Figure 4 represent the same data in a different format.

Lines 230-232: Are m <0 or >0?

Lines 246-280: Neither the meaning nor the origin of the parameters is explained.

3.1.1. This chapter does not contain any useful information.

Figure 5. Where did this structure come from?

3.2. This chapter does not contain any useful information.

Author Response

Dear Reviewer 2:

“The manuscript by Esteves et al. reports the solubility of picolinic acid in water, ethanol, and acetonitrile in the temperature range from 293.6 K to 323.4 K. These data are reported in Table 2 and Table 3.

The rest of the manuscript is a retelling of literature data, a series of muzzily obtained correlations, the main conclusion of which is the very common statement that " picolinic acid is more soluble the more polar the solvent", and powder diffraction experiments, which did not give any result. The structure is not very good, the research topic seems not substantiated and has nothing to do with crystals.

R: Dear Reviewer, this manuscript is the result of honest work executed by me and my colleagues. In fact, it started as the work of a student that was giving his first steps into research activities. I have conscience that it is not “hot”, however, in my opinion, its publication might be useful for some fellow scientist working in academia or in industry. Picolinic acid has many uses, for example, it is used to chelate zinc, and the zinc picolinate complex can be taken orally to help increasing this transition metal levels in the body. Picolinic acid is also an endogenous metabolite of tryptophan. Thus, having more data on solubility and crystallisation outcome of picolinic acid, in selected solvents at different temperatures, strikes me as useful, especially in water.

Additionally, we chose to embark on such simple studies due to the lack of understanding that still permeates the field of crystallization. Particularly in regard to events that take place at molecular level. With straightforward studies of small, structurally related, molecules, it will be possible to improve crystallisation processes. Bearing that in mind, we have been studying families of related molecules, such as hydroxynicotinic acids, and nicotinic acid and its isomers. Please note that picolinic acid is less explored in the literature in comparison to its isomers, nicotinic and isonicotinic acids. Hence, we have thought about contributing with new data, and with that, try to understand how the small differences in the structures of such small and simple molecules impacted both solubility and crystallisation. We would like to help establishing a benchmark for crystallisation processes, which could be fairly relevant for industry, particularly for the pharmaceutical one.

I sincerely hope that you might understand our reasons to publish this manuscript, and that you could be so kind and accept them.

The list of shortcomings:

The aim of the abstract is not to present the study but to summarize its most important results.

R: An alteration to the abstract was made following your feedback. Please find it in the revised version of the manuscript.

Figure 3 and Figure 4 represent the same data in a different format.

R: We kept both forms as they seemed useful to us. In this revised version we deleted Figure 3 to avoid repetition.

Lines 230-232: Are m <0 or >0?

R: The m values reported between lines 230 and 232 are from reference “Gonçalves, E.M.; Minas da Piedade, M.E. J. Chem. Thermodynamics 2012, 47, 362–371.” and are < 0. Thank you very much for noticing this error. We directly took the values from a table where they are positive, but the header of the table is −m, in fact there described as “−b”.

Lines 246-280: Neither the meaning nor the origin of the parameters is explained.

R: The parameters were used in Equation 2, which is basically the slope intercept formula, y = mx + b, where b represents the y value of the y intercept point and m the slope.

3.1.1. This chapter does not contain any useful information.

R: Dear Reviewer, in this section a systematisation of the information gathered in this work and the one available of this family of compounds was made. In our opinion, this might be useful so that other researchers can easily read and compare such information.

Figure 5. Where did this structure come from?

R: A small prismatic and colourless crystal, achieved by sublimation, using a petri dish.

3.2. This chapter does not contain any useful information.”

R: As picolinic acid is a relevant metabolite in the human body, accessing the acid-base behaviour of this compound, and once more systematise such information is, in our opinion, relevant for a fellow that might find need of such data. Only with the species distribution diagrams (Figure 9), calculated from the protonation constants (Table 7), can one predict the species present in aqueous solution at a given pH.

Reviewer 3 Report

In the paper entitled "Solubility and crystallization studies of picolinic acid", the authors investigate the solubility (using gravimetric methods) and solid-state structure of picolinic acid, a less studied isomer of nicotinic and isonicotinic acid. Applying different temperatures and three different solvents (water, ethanol and acetonitrile), the authors prove that picolinic acid is very soluble in water, and less soluble in the other two solvents used. Also, there were found two identifiable polymorphic forms for picolinic acid.

The paper is very well organized and correctly written and could be accepted for publication after a minor revision:

- page 5 - line 202: replace  "This work results" with "This work's results";

- I would advice the authors to replace the abbreviation used for acetonitrile (MeCN) with ACN, but this is a personal, yet not mandatory, request;

- The references cited are a bit old...a more up to date bibliography is advisable;

- in some of the references cited, the year of publication is not Bold faced.

Author Response

Dear Reviewer 3:

“In the paper entitled "Solubility and crystallization studies of picolinic acid", the authors investigate the solubility (using gravimetric methods) and solid-state structure of picolinic acid, a less studied isomer of nicotinic and isonicotinic acid. Applying different temperatures and three different solvents (water, ethanol and acetonitrile), the authors prove that picolinic acid is very soluble in water, and less soluble in the other two solvents used. Also, there were found two identifiable polymorphic forms for picolinic acid.

The paper is very well organized and correctly written and could be accepted for publication after a minor revision:

- page 5 - line 202: replace  "This work results" with "This work's results";

R: Dear Reviewer, thank you for the correction. This alteration is now on the revised version of the manuscript.

- I would advice the authors to replace the abbreviation used for acetonitrile (MeCN) with ACN, but this is a personal, yet not mandatory, request;

R: Dear Reviewer, if you do not mind, we prefer MeCN.

- The references cited are a bit old...a more up to date bibliography is advisable;

R: Please find more up-to-date literature in the revised version of the manuscript.

- in some of the references cited, the year of publication is not Bold faced.”

R: Dear Reviewer, in such cases the references are books.

Round 2

Reviewer 1 Report

Accepted

Author Response

Dear Reviewer,

thank you for accepting our manuscript. 

All the best.

Reviewer 2 Report

Regardless of whether the article is written by an expert or a student, it must meet the mandatory requirements.

The abstract should summarize the purpose and the most important results of the work and be understandable without reading the body of the article.

“Crystallization is a method to obtain solids from solution that has been around for a long time. However, it remains unclear how molecules form a given crystal. Variables such as temperature and solvent used can affect solubility. The crystallization outcome can also be affected, thus polymorphism appearance must be investigated”. This article is not a review of crystallization.

“(PA, Figure 1)” - -the abstract must not refer to the article.

Picolinic acid is not an isomer of isonicotinic and nicotinic acids. All three acids are isomers of pyridinecarboxylic acid.

The reader should be able to understand the origin of the reported data.

The parameters presented in Table 4 refer to 298 K. Give a mathematical description of how the temperature dependence of the solubility can be calculated based on these parameters (Table 5).

Who measured the single crystal structure shown in Figure 4?

Reported data should be relevant to the research topic.

3.2. Acid–base behavior of picolinic, nicotinic and isonicotinic acids, speciation in water, and its influence on their solid-state structures. How is this part related to the solubility or crystallization of PA?

Author Response

“Regardless of whether the article is written by an expert or a student, it must meet the mandatory requirements.”

R: Dear Reviewer, I agree with you.

“The abstract should summarize the purpose and the most important results of the work and be understandable without reading the body of the article.”

R: Thank you for your input. I tried to enhance it again. Please find the new revisions in the manuscript revised file.

““Crystallization is a method to obtain solids from solution that has been around for a long time. However, it remains unclear how molecules form a given crystal. Variables such as temperature and solvent used can affect solubility. The crystallization outcome can also be affected, thus polymorphism appearance must be investigated”. This article is not a review of crystallization.”

R: Dear Reviewer, thank you for your comment.

“(PA, Figure 1)” - -the abstract must not refer to the article.”

R: This was removed.

“Picolinic acid is not an isomer of isonicotinic and nicotinic acids. All three acids are isomers of pyridinecarboxylic acid.”

R: Dear Reviewer, this was amended throughout the manuscript.

“The reader should be able to understand the origin of the reported data.”

R: Dear Reviewer, I agree with you.

“The parameters presented in Table 4 refer to 298 K. Give a mathematical description of how the temperature dependence of the solubility can be calculated based on these parameters (Table 5).”

R: The results presented in Table 5 were obtained by linear regressions, using the least squares method, of ln xPA and each parameter. Please note that these parameters are temperature dependent, thus, in Table 4 the temperature value used is referred (298K).

“Who measured the single crystal structure shown in Figure 4?”

R: Professor M. Fátima M. Piedade.

“Reported data should be relevant to the research topic.”

R: Dear Reviewer, once again, I agree with you.

"3.2. Acid–base behavior of picolinic, nicotinic and isonicotinic acids, speciation in water, and its influence on their solid-state structures. How is this part related to the solubility or crystallization of PA?”

R: Dear Reviewer, in my humble opinion, this part is related to both solubility and crystallization of PA. With the analysis of the acid–base behavior of picolinic acid and its speciation in water, not only a reason for its increased solubility in this solvent could be uttered, but also the influence it plays on its solid-state structure expressed. PA is much more soluble in water than expected and the fact that it has a large pH interval for the abundance of its AH species, correlates well with its increased solubility in water. Please note that protonation constants are fundamental to rationalize about the diverse behavior of each compound in aqueous solution. And to know which species might be present at a given pH can help predicting the crystallization outcome.

Thanks for all the time and attention given to this work, which nowadays is increasingly becoming a rare thing to find.

Round 3

Reviewer 2 Report

I have no further comments to make regarding this manuscript.